# Remote Sensing Scene Data Generation Using Element Geometric Transformation and GAN-Based Texture Synthesis

Zhaoyang Liu [1], Renxiang Guan [1], Jingyu Hu [1], Weitao Chen [1,2] and Xianju Li [1,2,*]

1    School of Computer Science, China University of Geosciences, Wuhan 430074, China; liuzhaoyang9562@cug.edu.cn (Z.L.); cugjsjkjb@163.com (R.G.); hhyuyu2000@163.com (J.H.); wtchen@cug.edu.cn (W.C.)
2    Hubei Key Laboratory of Intelligent Geo-Information Processing, China University of Geosciences, Wuhan 430074, China
*   Correspondence: ddwhlxj@cug.edu.cn

**Abstract:** Classification of remote sensing scene image (RSSI) has been broadly applied and has attracted increasing attention. However, scene classification methods based on convolutional neural networks (CNNs) require a large number of manually labeled samples as training data, which is time-consuming and costly. Therefore, generating labeled data becomes a practical approach. However, conventional scene generation based on generative adversarial networks (GANs) involve some significant limitations, such as distortion and limited size. To solve the mentioned problems, herein, we propose a method of RSSI generation using element geometric transformation and GAN-based texture synthesis. Firstly, we segment the RSSI, extracting the element information in the RSSI. Then, we perform geometric transformations on the elements and extract the texture information in them. After that, we use the GAN-based method to model and generate the texture. Finally, we fuse the transformed elements with the generated texture to obtain the generated RSSI. The geometric transformation increases the complexity of the scene. The GAN-based texture synthesis ensures the generated scene image is not distorted. Experimental results demonstrate that the RSSI generated by our method achieved a better visual effect than a GAN model. In addition, the performance of CNN classifiers was reduced by 0.44–3.41% on the enhanced data set, which is partly attributed to the complexity of the generated samples. The proposed method was able to generate diverse scene data with sufficient fidelity under conditions of small sample size and solve the accuracy saturation issues of the public scene data sets.

**Keywords:** GAN; CNN; data generation; texture synthesis; classification

## 1. Introduction

With the continuous development of earth observation technologies, increasing numbers of remote sensing satellites have been launched into orbit, which enabled the collection of a myriad of high-resolution remote sensing (HRRS) image data. High-resolution remote sensing images not only exhibit characteristics of rich spectral features, various shapes, and abundant texture features, but also contain significantly clear semantic scene information. Therefore, the classification of remote sensing image scenes is a significant approach for analyzing remote sensing image information, which has attracted considerable research efforts as a key technology for interpreting images and understanding the real world. The classification of high-resolution remote sensing scene data is the basis of several major applications of remote sensing, such as land resource management, natural resource management, and urban planning. Generally, remote sensing scene images (RSSI) reflect widely varying types of objects (including landforms in distinct conditions and buildings with diverse shapes) with complicated features. Hence, the classification of such information remains challenging.

The classification of remote sensing images is hugely dependent on data labeling, in which humans input meaningful labeled data into the training model, and subsequently, such data is further used in feature modeling. However, at present, in the context of expanding applications of such data, the lack of existing pre-labeled remote sensing data becomes a significant obstacle to remote sensing scene classification research, requiring the expenditure of considerable labor and time. Thus, automated generation of marked remote sensing scene data has become practical.

At present, existing methods of data generation have been mainly developed based on generative adversarial networks (GANs). Problems such as distortion and long training times have been reported in the generation of remote sensing scene data using traditional GAN networks and associated variants. Karras et al. [1] proposed a gradually developed generator and discriminator GAN to generate natural images such as human faces and animals. However, the size of the generated images was limited. Subsequently, it was found that this method could not be applied to RSSIs. Zhao et al. [2] used pre-trained GAN models to generate natural images under conditions of extremely limited data. This is consistent with the lack of scene data addressed in the present work. However, after experiments, it was found that problems such as distortion persisted. Han et al. [3] proposed a high-resolution remote sensing scene data generation method based on a Wasserstein GAN network, which solved the problem of distortion to a certain extent. However, the size of the generated data was limited. Pan et al. [4] proposed a diversified improved GAN with the ability to generate RSSIs. However, due to limitations in computing power and training time, this method nonetheless was unable to learn to generate scene images of sufficient size.

Although generating RSSIs directly is evidently difficult for GAN networks, they perform effectively with respect to texture synthesis tasks. At present, several texture analysis methods have been developed. Among the statistical analysis methods, Sklansky [5] proposed and applied an approach to describe texture features through autocorrelation function. Weszka et al. [6] proposed a gray difference histogram statistical method, which was able to describe the spatial organization information of grayscale images. Methods based on the characteristics of texture primitives and their arrangement rules are known as structural analysis methods. For example, Tuceryan and Jain [7] used this structural analysis method and put forward texture segmentation depending on the Voronoi polygon. Texture analysis based on models assumes that the texture is distributed according to a given model. Spectrum analysis methods are constructed on the multi-scale analysis of texture. For example, Fergusson and Gabor [8] proposed the Gabor function [9] as a windowed Fourier-transform method. This approach addressed the deficiency that the classical Fourier transform cannot analyze time and frequency simultaneously. Based on a Markov random field to generate specific texture images and the analysis of image texture features, Efros and Leung [10] developed a method that could repair a missing image successfully. Zhou et al. [11] used a GAN to synthesize large-area images with sample texture features based on given small sample textures. Jetchev et al. [12] proposed a new texture synthesis model based on GAN learning.

In the present work, by extending the input noise distribution space from a single vector to an entire space tensor, a framework with properties that are well-suited for texture synthesis tasks is created, called spatial GAN (SGAN). To the best of our knowledge, the proposed method is the first successful GAN-based texture synthesis method that is completely data-driven.

A typical RSSI can be divided into two parts. For example, it can be considered that a lake scene image contains two parts. One part represents the element of the scene category, which is the lake in this case, and the other part is the element other than the above, which we refer to as the background. In a lake scene, the background might be grassland, desert, or mountains. First, we apply k-means clustering [13] to perform scene image segmentation. To address the noise of the image after segmentation, the proposed approach then extracts the elements of the scene geometry characteristic information and

translates them into a homogeneous coordinate matrix to apply the techniques originating in computer graphics. After using a simple and complex geometry transform, a complex scene structure is obtained. Simultaneously, we sample the texture features of the elements and use texture synthesis technology to synthesize a large amount of texture feature data based on the collected texture samples. Finally, we fill the texture information back into the reconstructed scene structure, resulting in a new, high-fidelity scene image of increased complexity.

The remainder of this study is organized as follows. In Section 2 below, we introduce the proposed method, while the results of the experiments are provided in Section 3. Sections 4 and 5 present our discussion and conclusions and suggests some possible avenues for future research.

## 2. Materials and Methods

The proposed RSSI generation exhibits a close relationship with the human perception of remote sensing image scenes. Primarily, we divided an RSSI into two parts. One is used to cause the scene image that represents the feature elements of the corresponding scene image category, such as lakes, rivers, islands, etc., while the other part is to cause them to be recognized as the background, except for natural features such as the grassland around the lake, the ocean surrounding the island, etc.

First, we reconstructed the combined feature structure relationship among scene elements through a random transformation of the elements and created a scene element system with diverse and complex structures. Subsequently, we synthesized the texture features of the background with a depth generation algorithm, in order to obtain a scene image with random, complex, and authentic geometry and texture. Compared with the directly generated image, this method uses the depth generation model to construct the spectral information of pixels, which involves a relatively low possibility of distortion.

Figure 1 shows a general overview of the technical roadmap of our method. First, we extracted the geometric polygons of the elements in the original scene image via image segmentation, and subsequently, performed geometric transformations on the obtained polygons through computer graphics. Finally, we used texture synthesis technology to synthesize the texture of the corresponding ground object, and subsequently, merged it into a new RSSI.

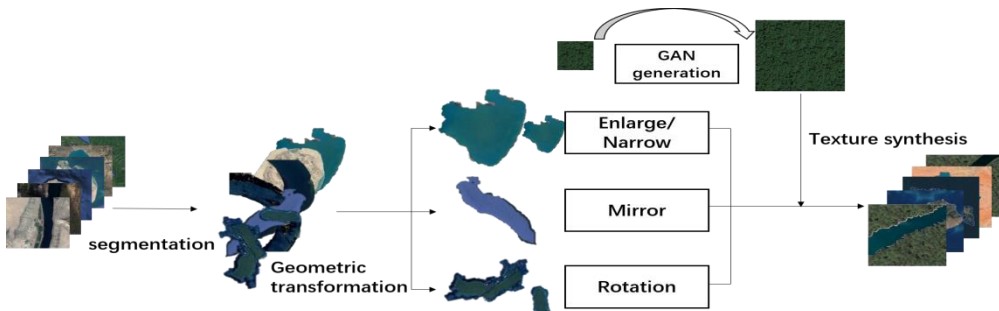

**Figure 1.** An overview of our proposed method.

### 2.1. Extraction of Remote Sensing Scene Image Elements Based on k-Means Clustering

The k-means clustering algorithm is a conventional clustering method that uses space efficiently. Clustering methods divide data into several groups. Based on the defined measurement standard, the data in a given group is more similar to other groups of data. The calculation of similarity involves detecting the distance between the data object and the cluster center. The closest distance to the cluster center is divided among the cluster. The procedure is given as follows. First, randomly select k objects that initially represent the average or center of the cluster. Each other object is allocated to the nearest cluster according to its distance from the center of each cluster. Then, calculate the average value of each cluster again to find the center of the cluster, and reorder the clusters accordingly.

This process is repeated until the criterion function converges. The time complexity of this algorithm is $O(nkt)$, where $n$ is the number of all objects, $k$ is the number of clusters, and $t$ is the number of iterations.

In this study, we used a k-means clustering method to segment remotely sensed scene images. We inputted the image as a matrix into the algorithm, specified the parameter k for clustering (that is, segmentation) and finally, achieved the purpose of extracting elements in the scene.

The proposed method uses the k-means clustering method to segment scene images. A set of RSSIs is input into the algorithm, and subsequently, the separated images are obtained. However, these images still cannot be used to extract the elements in the RSSIs. This is because both the background part and the element part that are divided contain some noise data, which are expressed as dense or washed-out blobs or obstructions in the image. Therefore, to extract the elements through the segmented image, we must first preprocess it.

*2.2. Element Geometric Transformation*

After the image segmentation, we obtained the geometric features of each element in the corresponding scene image. These geometric features can be used to demonstrate the structural relationship between various elements in RSSIs and to generate RSSIs.

In order to generate a new RSSI that is distinct from the original RSSIs, it is necessary to transform the geometric features of the extracted elements for the purpose of reconstructing the entire RSSI. The transformation that we performed involves enlargement, reduction, translation, and symmetry (on both X- and Y-axes). To complete the mathematical description of feature transformation, we first abstracted the extracted feature as a polygon. We used a homogeneous coordinate matrix P to record the coordinates of each vertex counterclockwise from a vertex of the polygon. Assuming the coordinates of each vertex of a polygon to be $(x_0, y_0)\,(x_1, y_1) \ldots (x_n, y_n)$, respectively, in the counterclockwise direction, the corresponding homogeneous coordinate matrix is given by

$$\begin{pmatrix} x_0 & y_0 & 1 \\ x_1 & y_1 & 1 \\ \ldots & \ldots & \ldots \\ x_n & y_n & 1 \end{pmatrix}. \tag{1}$$

Subsequently, we defined a $3 \times 3$ geometric transformation matrix $T$, which is used for geometric transformation of polygons. The transformation matrices of the geometric transformation were implemented as follows.

Enlargement and reduction were given by

$$T = \begin{pmatrix} S_x & 0 & 0 \\ 0 & S_y & 0 \\ 0 & 0 & 1 \end{pmatrix} \tag{2}$$

where $S_x$ and $S_y$ represent the proportion of enlargement and reduction. When $S_x$ and $S_y$ are greater than 1, the operation is enlargement. When $S_x$ and $S_y$ are less than 1, the operation is reduction.

The translation operation was performed as follows.

$$T = \begin{pmatrix} 1 & 0 & 0 \\ 0 & 1 & 0 \\ T_x & T_y & 1 \end{pmatrix}, \tag{3}$$

where $T_x$ and $T_y$ represent the translation distance of features in $x$ and $y$ directions.

Counterclockwise rotation was performed as follows.

$$T = \begin{pmatrix} \cos\theta & \sin\theta & 0 \\ -\sin\theta & \cos\theta & 0 \\ 0 & 0 & 1 \end{pmatrix} \tag{4}$$

where $\theta$ is the counterclockwise rotation angle of the element.

Clockwise rotation was performed as follows.

$$T = \begin{pmatrix} \cos(-\theta) & \sin(-\theta) & 0 \\ -\sin(-\theta) & \cos(-\theta) & 0 \\ 0 & 0 & 1 \end{pmatrix}, \tag{5}$$

where $\theta$ is the counterclockwise rotation angle of the element.

The *X*-axis symmetry transformation is given by

$$T = \begin{pmatrix} 1 & 0 & 0 \\ 0 & -1 & 0 \\ 0 & 0 & 1 \end{pmatrix}, \tag{6}$$

While *Y*-axis symmetry is given by

$$T = \begin{pmatrix} -1 & 0 & 0 \\ 0 & 1 & 0 \\ 0 & 0 & 1 \end{pmatrix}, \tag{7}$$

Assuming that the coordinates of each vertex of the transformed polygon are $(x_0', y_0')(x_1', y_1')\ldots\ldots\ldots(x_n', y_n')$, consequently, the homogeneous coordinate matrix P composed of them is given by

$$\begin{pmatrix} x_0' & y_n' & 1 \\ y_0' & y_1' & 1 \\ \ldots & \ldots & \ldots \\ x_n' & y_n' & 1 \end{pmatrix}. \tag{8}$$

After we perform these basic geometric transformations, we can compound these basic geometric transformations, and the corresponding geometric transformation formula is given by

$$P' = PT_1 T_2 \ldots \tag{9}$$

Then, $T_i$ is used to choose the corresponding transformation matrix according to the transformation needed. The significance of the formula is that the geometric transformation represented by $T_1$ matrix is carried out first, and on this basis, the geometric transformation represented by $T_2$ matrix is carried out, and so on.

After the transformation, we could obtain the new structure of the reconstructed RSSI, which lays the foundation for the next section.

### 2.3. Texture Synthesis of RSSI Based on GAN Network

In a typical GAN network [14], the generator network maps randomly generated noise data, such as Gaussian noise, to fake samples. The discriminator network must receive real data samples (to discriminate between data samples) or fake samples. The generator is then trained to cheat the discriminator. Simultaneously, the discriminator also needs to be trained to be able to overcome the generator. Generally, the competition between the generator and the discriminator enables the generator to generate enough samples to deceive the discriminator, and the discriminator can better distinguish between true

samples and constructed samples. The min-max objective function of the game between the generator $G$ and the discriminator $D$ is given by

$$\min_G \max_D E_{x \sim Pr}[\log(D(x))] - E_{\widetilde{X} \sim P_g}[\log(D(\widetilde{x}))], \tag{10}$$

where $P_r$, $x$ and $P_g$ are the real data distribution, real sample and model distribution, respectively. Fake samples can be represented as $\widetilde{x} = G(z)$, $Z = P(z)$, where the input $z$ of the generator is sampled from a simple noise distribution, such as a spherical Gaussian distribution. If the optimization and training of the discriminator are after the generator updates the parameters, then the minimization of the $JS$ divergence between $P_r$ and $P_g$ is the minimization value function. $JS$ divergence is an important indicator of the similarity between two probability distributions based on Kullback–Leibler ($KL$) divergence. $JS(P_r, P_g)$ in GAN can be calculated as follows.

$$JS(P_r, P_g) = KL(P_r, P_m) + KL(P_g, P_m) \tag{11}$$

where $KL(P_g, P_m)$ is defined as $\int \log\left(\frac{P_R(x)}{P_g(x)}\right) P_r(x) d\mu(x)$ Both $P_r$ and $P_g$ are assumed to be absolutely continuous and admit densities.

Based on the work of Zhou et al. and Efros et al. on texture synthesis, we used a GAN to synthesize the texture of each element extracted. The texture synthesis technology in this section is based on the work of Efros and Zhou. We sampled the surface texture of rivers, forests, grasslands, and oceans in RSSIs and selected a big number of typical surface texture samples as the input of texture synthesis. When performing texture synthesis, we selected a total of 100 texture samples from five categories, including rivers, lakes, seas, meadows, and forests, representing the typical characteristics of the corresponding category. These five classes of samples were taken from rivers, lakes, ocean forests, and grasslands. Figure 2 is an example of our selection of samples.

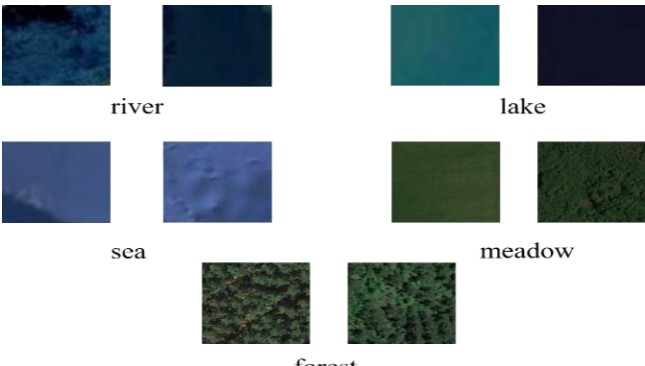

**Figure 2.** The texture samples from five classes in the data set.

### 2.4. Scene Data Generation Quality Assessment

After obtaining the generated scene images, we evaluated the classification accuracy of the generated images. Firstly, a visual discrimination method was used to judge the generated images with the naked eye, and they were compared with the images directly generated by other generated adversarial networks. Secondly, the three conventional convolutional neural networks (CNN)-ALEX-NET [15], VGG-NET [16], and RES-NET [17] were used to carry out classification experiments on the data sets of NWPU-RESISC45 [18]. Then, the generated scene images were classified. In total, 60 images from each of the 5 categories were added to the original data set to form an enhanced data set, and classification experiments were carried out on the enhanced data set. In addition, we also conducted a separate classification experiment on the scene images generated by the proposed approach.

## 3. Results

This section introduces the data set used in the experiment and shows and compares the scene images generated by some mainstream GAN networks and the scene images using our method. Finally, we compared the proposed method using conventional CNNs for classification of the generated images to test the efficacy of the proposed approach.

### 3.1. Data Set Description and Experiment Setting

We used the NWPU-RESISC45 public HRRS data set to evaluate our method.

The NWPU-RESISC45 data set contained 45 scene categories with 700 images per category, for a total of 31,500 images. Each image included color with a size of 256 × 256 pixels in an RGB color space. Figure 3 shows some of the sample images we used in the data set. This data set is produced by experts in the field of remote sensing based on Google Earth, and its spatial resolution varied from 0.2 m to 30 m per pixel. It involved viewpoint, object attitude, background, occlusion, translation, and other changes, and had significant inter-class similarity and intra-class diversity. Hence, the data set is suitable for the development and evaluation of a variety of networks.

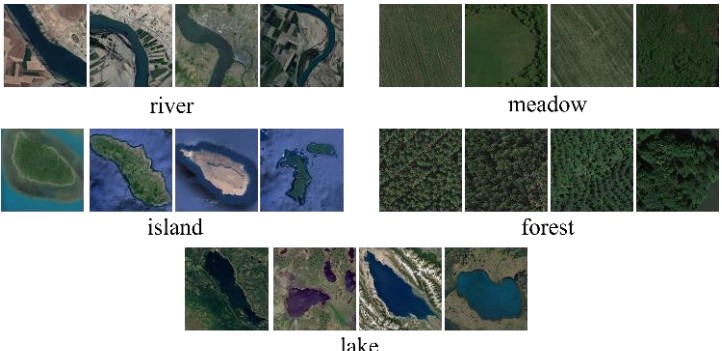

**Figure 3.** Examples from the NWPU-RESISC45 data set. (Each class shows four images).

### 3.2. Results of Segmentation, Element Geometric Transformation and Texture Synthesis

Four examples are shown in Figure 4 of the results of segmentation on a lake, river, and island. In each example, the original image is shown on the left, and in the middle, the k-means segmentation results are shown, and the final processed segmented result is on the right.

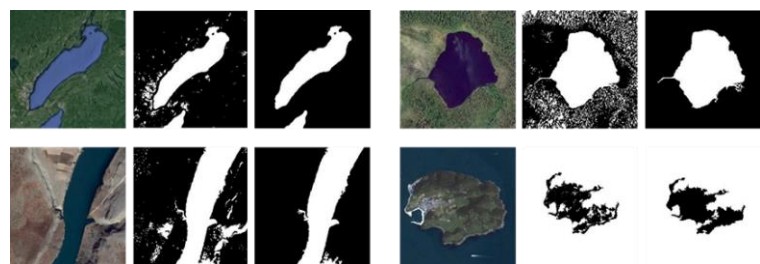

**Figure 4.** Results of segmentation after processing on lake, river and island.

After the geometric boundary of the elements was obtained, we carried out a geometric transformation on the geometric boundary of the elements. In Figure 5, the leftmost side of each line is the original scene image, and then the geometric transformation of zooming in, shrinking, translation, and rotation is shown successively from left to right.

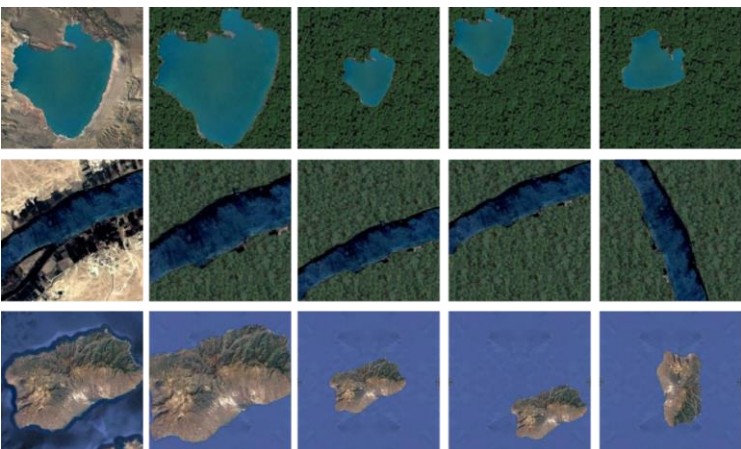

**Figure 5.** Results of geometric transformation on elements of lake, river, and island.

The results of texture synthesis are as follows. Four examples are shown in Figure 6.

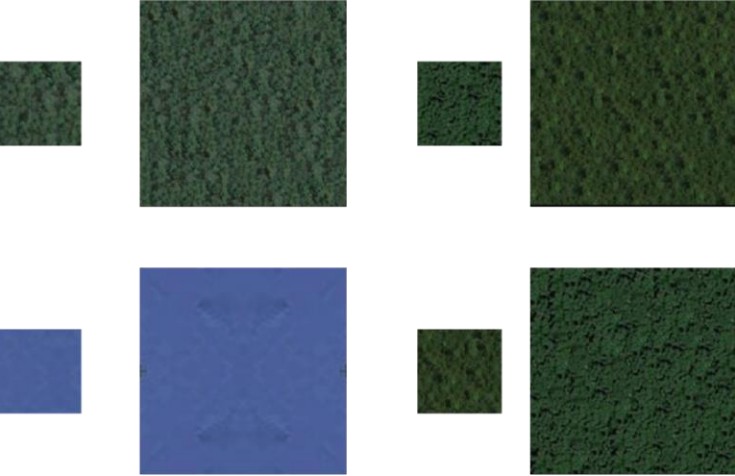

**Figure 6.** Results of texture synthesis; the smaller ones are samples selected from the data set and the bigger ones are the synthesized texture images.

*3.3. Final Results of Filling the Element Polygon with the Textures and Comparison of the Results of GANs*

In this Section, we provide the results of tests conducted to examine the effectiveness of conventional GAN network models in RSSI generation and compared them with our proposed approach.

Figure 7 shows the comparison of the effects of RSSI samples generated by mainstream GAN networks and by our proposed method.

We compared images generated by our method with the image generated by the conventional GAN network. It may be observed that for most images generated by the GAN network, there are problems with blur and distortion. In the ACGAN network generation experiment, there was a significant gradient-vanishing issue, and the model did not achieve a degree of convergence that fulfilled the requirements. The image generated by WGAN did not have enough detailed information. The effect of the image generated by other GAN networks was better than the former two, but the size of the generated image was strictly limited. It required more time to train an image of the same size as the original data set, and these times were often excessive. The RSSIs generated by our proposed method were not only higher in resolution than the images generated by the GAN network, but also our method was able to quickly generate an image of any size.

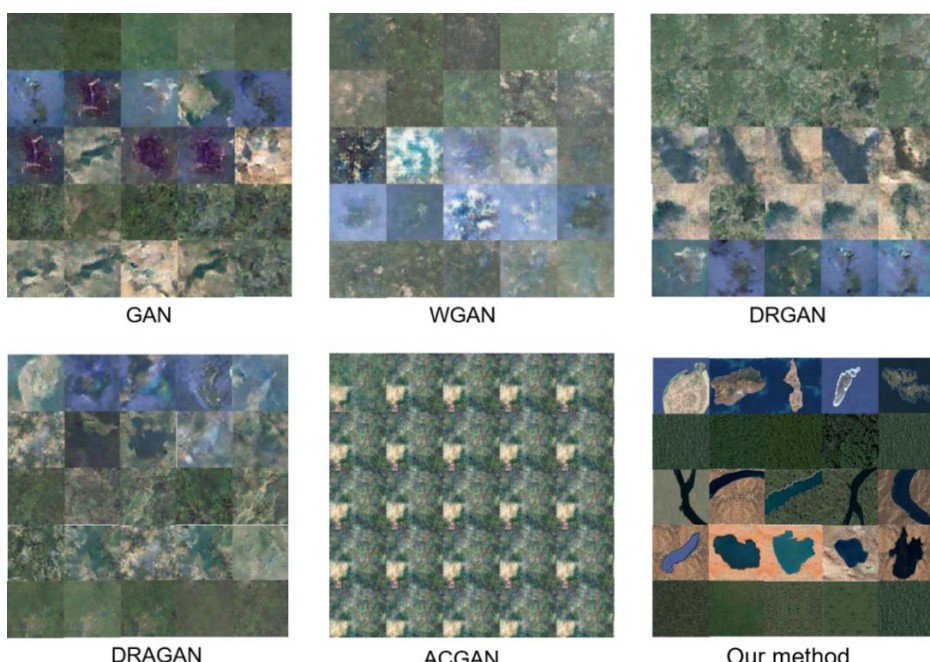

**Figure 7.** Generation results of GANs and our method.

After extracting and transforming the geometric boundaries of the elements in the scene image, we used the synthetic texture to fill in the geometric contours of the elements and obtained the following results.

### 3.4. Classification Accuracy Evaluation

3.4.1. Data Set Split and Experiment Settings

All experiments were conducted on NVIDIA workstations. The specific experimental setup and the deep learning framework used are shown in Table 1.

**Table 1.** Workstation configurations.

| Hardware Platform | Parameter Name |
|---|---|
| CPU | E5-2068 v3 (2.50 GHz 2 cores) |
| Memory | 256 GB |
| GPU | NVIDIA GeForce RTX 2080 Ti |
| Memory | 11.0 GB |
| Software Platforms | Pycharm 2020.03 |
| OS | Windows 10 |
| DL Framework | Pytorch |
| CUDA | 11.1 |

In the two sets of experiments classified by CNNs, we divided the data set and investigated the changes in classification accuracy under different training set proportions. The settings of data set are represented in Table 2. And Table 3 shows the number of various geometric transformation in each class. In order to ensure the reliability of the data, each set of experiments was repeated three times.

**Table 2.** The settings of the data set.

| Data Set Setting | Training Set Ratio | Test Set Ratio | No. Per Class for NWPU 45 | |
|---|---|---|---|---|
| | | | Training | Test |
| (1) | 20% | 80% | 140 | 560 |
| (2) | 40% | 60% | 280 | 420 |
| (3) | 60% | 40% | 420 | 280 |

**Table 3.** The number of various geometric transformation in each class.

| Name of the Class | Rotation | Enlarge | Narrow | Panning |
|---|---|---|---|---|
| Island | 15 | 20 | 17 | 8 |
| River | 32 | 5 | 5 | 18 |
| Lake | 20 | 15 | 15 | 10 |
| Forest | ... | ... | ... | ... |
| Meadow | ... | ... | ... | ... |

3.4.2. Experiment 2: Evaluation of Classification Capacity

This section introduces the three conventional CNNs used for comparison, including VGG-Net, Res-Net, and Alex-Net. The performance of the original data set and the enhanced data set after adding samples generated using our method was compared, and we used this to investigate the impact of the generated data on the accuracy of the networks.

First, we fixed the original data set and increased the number of generated samples as an enhanced data set. On five categories of the NWPU-RESISC45 public HRRS data set, we tested rivers, grasslands, forests, lakes, and islands to 60 pieces of data collected for each category, and the number of training rounds, training set ratio, etc., remained unchanged. The only variable was the size of the data set.

First, we performed a classification experiment on the original data set, using 20%, 40%, and 60% of the original data set as a training set for the experiment. After obtaining the classification accuracy, we added the generated data, keeping the size of the training set consistent with the previous experiment, and performed the classification experiment to obtain the classification accuracy.

In the experiment evaluating the quality of generated data quality evaluation, the training settings of the three CNNs we used are as follows. The learning rate was 0.001, the batch size was 64, the loss function was BCE, and we used the stochastic gradient descent optimizer to update the CNN model.

Table 4 shows the accuracy results for the original NWPU-RESISC45 data set. Table 5 shows those on the NWPU-RESISC45 data set after adding data generated by our method. Tables 6 and 7 shows those on the data generated by our approach. Table 6 is the result of data size 64 × 64 and Table 7 is the result of data size of 256 × 256.

**Table 4.** Accuracy results for the NWPU-RESISC45 data sets.

| Network | 20% Training Set | 40% Training Set | 80% Training Set |
|---|---|---|---|
| AlexNet | $84.02 \pm 1.52\%$ | $88.53 \pm 3.02\%$ | $89.44 \pm 1.18\%$ |
| VGGNet | $83.11 \pm 1.33\%$ | $90.83 \pm 2.11\%$ | $90.34 \pm 1.11\%$ |
| ResNet | $95.71 \pm 0.72\%$ | $96.75 \pm 0.16\%$ | $97.82 \pm 0.45\%$ |

**Table 5.** Accuracy results for the NWPU-RESISC45 data sets after adding data generated by our method.

| Network | 20% Training Set | 40% Training Set | 80% Training Set |
|---|---|---|---|
| AlexNet | $80.61 \pm 1.56\%$ | $86.57 \pm 2.72\%$ | $88.71 \pm 1.52\%$ |
| VGGNet | $81.68 \pm 1.41\%$ | $84.73 \pm 1.98\%$ | $91.43 \pm 1.85\%$ |
| ResNet | $95.21 \pm 0.68\%$ | $96.31 \pm 0.32\%$ | $96.97 \pm 0.55\%$ |

**Table 6.** Accuracy results for the data generated by our approach (64 × 64).

| Network | 20% Training Set | 40% Training Set | 80% Training Set |
|---|---|---|---|
| AlexNet | $72.95 \pm 5.50\%$ | $83.22\% \pm 2.03\%$ | $57.86 \pm 7.56\%$ |
| VGGNet | $77.53 \pm 3.52\%$ | $78.77\% \pm 3.52\%$ | $81.00 \pm 4.38\%$ |
| ResNet | $94.23 \pm 3.01\%$ | $98.30\% \pm 1.62\%$ | $98.45 \pm 1.62\%$ |

**Table 7.** Accuracy results for the data generated by our approach ($256 \times 256$).

| Network | 20% Training Set | 40% Training Set | 80% Training Set |
|---|---|---|---|
| AlexNet | $70.42 \pm 4.77\%$ | $80.50 \pm 2.86\%$ | $82.30 \pm 3.01\%$ |
| VGGNet | $53.88 \pm 9.06\%$ | $80.44 \pm 2.36\%$ | $79.62 \pm 7.00\%$ |
| ResNet | $95.23 \pm 0.62\%$ | $99.21 \pm 0.82\%$ | $100.00 \pm 0.00\%$ |

It may be observed that all of the conventional models used for comparison exhibited decreased training accuracy when the number of training sets was 20% and 40% of the original data set, and the range of the decrease was between 0.44–3.41%, while in the training set when the number was 60%, the training accuracy of Alex-Net and ResNet also decreased, and the range of the decrease range was between 0.73% and 0.85%. In the classification experiment performed only on our own generated data, it may be observed that almost all classification accuracy results were less than the classification accuracy of the original data set.

## 4. Discussion

We conducted extensive experiments on the public NWPU-REISC45 HRRS data set to justify the effectiveness and feasibility of the proposed method and obtained the expected results. As for the classification ability of CNNs on the scene images, it was found that the classification accuracy of all the methods was reduced. For the reduction in the accuracy in the comparison experiments between the original data set and the enhanced data set, we think that after adding the data generated by our proposed method, the complexity of the data set increased and the separability decreased. Additionally, for the classification experiment of the generated data, we think that this is because the amount of data used in this experiment is small, which leads to insufficient model training. All in all, the complexity of the enhanced data set was increased, and our proposed method partially solves the accuracy saturation issues of those public scene data sets.

With respect to the classes for data generation, we selected the classes that contain elements that are more blocky and whose textures are smooth. This means it is difficult to apply to those classes that do not meet the requirements above.

While performing the data generation experiments, we noticed that ethical awareness is of great importance to concern. With the artificial intelligence, especially the deep fake technology so developed, great convenience has been brought to us. Meanwhile, these technologies have also raised great ethical concerns. The generation and recognition technology of face images has caused the violation of privacy and the theft of personal information. In the field of remote sensing images, law enforcement of satellite images and land use approval are also facing similar challenges. The ethical issues of artificial intelligence in the field of remote sensing still need careful research and thinking.

## 5. Conclusions

In this study, to solve the problems of small sample size in a remote sensing scene data set, as well as simple scene structure and distortion caused by direct generation, a scene reconstruction method was proposed based on random element transformation and GAN-based texture synthesis. This method can be applied to generate remote sensing scene data set with labels, in which the random transformation of elements can be used to reconstruct the scene structure and the texture synthesis can be used to fill the spectrum of the reconstructed scene. Experimental results demonstrate that the RSSI generated by our method achieved a better visual effect than a GAN model. In addition, the performance of CNN classifiers was reduced on the enhanced data set, which is partly attributed to the complexity of the generated samples. Our work can generate a large number of scene images with complex structures and high fidelity. Thus, the existing remote sensing scene data sets can be expanded, alleviating the accuracy saturation issues. In addition, pre-

trained classification models for RSSIs can also be constructed based on the enhanced data sets.

In the future, we expect to generate more images with complex scene structures and realistic descriptions, to promote the development of scene image generation in remote sensing.

**Author Contributions:** Conceptualization, Z.L. and X.L.; methodology, Z.L.; validation, R.G. and J.H.; investigation, Z.L.; resources, X.L.; writing—original draft preparation, Z.L.; writing—review and editing, X.L. and W.C.; visualization, Z.L.; supervision, X.L.; project administration, X.L.; funding acquisition, X.L. All authors have read and agreed to the published version of the manuscript.

**Funding:** This work was supported by Natural Science Foundation of China (No. 42071430) and College Students' Innovative Entrepreneurial Training Plan Program (No. 202010491040).

**Data Availability Statement:** The NWPU-RESISC45 data set can be obtained through http://www.escience.cn/people/JunweiHan/NWPU-RESISC45.html, accessed on 24 February 2022.

**Conflicts of Interest:** The authors declare no conflict of interest.

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
