# Peer review of "Remote Sensing Scene Data Generation Using Element Geometric Transformation and GAN-Based Texture Synthesis"

_applsci, doi:10.3390/app12083972_

Round 1
Reviewer 1 Report
While is clearly explained, in the proposed article, that the classification of high-resolution remote sensing scene data is the basis of land resource management, natural resource management, urban planning, etc., the applicability of the RSSI generated by a synthetic process is not specified.
Ethical awareness for artificial intelligence should be discussed in the proposed article. Once specified the possible fields of application, please, address the possible ethical issues that the usage of synthetic instances of data (that can pass for real data to the human eye and to the machine itself) could generate in real life.
Typo
Line 257: Berkeley not in bibliography
Line 341: also repeated twice
Line 346-350: All the introduction text in the "Discussion" section is taken from the Template provided by the editors. Please, remove/replace it.
Suggestions:
Figure 3. Classes description too small (river, meadwod, island, forest, lake)
Reviewer 2 Report
The work exposes a method of generating scene data using GAN to augment the data sets needed for remote image classification algorithms. The objectives and the proposed method are very clearly indicated. It is recommended to explain more consistently the results and the advantages that are achieved. There are no improvements in the accuracy of the classification, as is the case in the references provided by the authors.
Line 253 should be corrected as the equation changes the format.
On line 416 the name of the dataset used should be in uppercase as in the rest of the document.
Round 2
Reviewer 1 Report
Misunderstanding at Point 6: Figure 3. Classes description too small (river, meadow, island, forest, lake): please note the reviewer meant that the font size of the text (river, meadow, island, forest, lake) on the images in Figure 3 is too small and unreadable. Please increase font size.
